# Rethinking Soft Labels for Knowledge Distillation: A Bias-Variance Tradeoff Perspective

**Helong Zhou**[1*], **Liangchen Song**[2*†], **Jiajie Chen**[1*], **Ye Zhou**[1], **Guoli Wang**[13],
**Junsong Yuan**[2], **Qian Zhang**[1]
[1]Horizon Robotics    [2]University at Buffalo    [3]Tsinghua University
{helong.zhou,jiajie.chen,ye.zhou,guoli.wang}@horizon.ai
{lsong8,jsyuan}@buffalo.edu, qian01.zhang@horizon.ai

## Abstract

Knowledge distillation is an effective approach to leverage a well-trained network or an ensemble of them, named as the teacher, to guide the training of a student network. The outputs from the teacher network are used as soft labels for supervising the training of a new network. Recent studies (Müller et al., 2019; Yuan et al., 2020) revealed an intriguing property of the soft labels that making labels soft serves as a good regularization to the student network. From the perspective of statistical learning, regularization aims to reduce the variance, however how bias and variance change is not clear for training with soft labels. In this paper, we investigate the bias-variance tradeoff brought by distillation with soft labels. Specifically, we observe that during training the bias-variance tradeoff varies sample-wisely. Further, under the same distillation temperature setting, we observe that the distillation performance is negatively associated with the number of some specific samples, which are named as regularization samples since these samples lead to bias increasing and variance decreasing. Nevertheless, we empirically find that completely filtering out regularization samples also deteriorates distillation performance. Our discoveries inspired us to propose the novel weighted soft labels to help the network adaptively handle the sample-wise bias-variance tradeoff. Experiments on standard evaluation benchmarks validate the effectiveness of our method. Our code is available at https://github.com/bellymonster/Weighted-Soft-Label-Distillation.

## 1 Introduction

For deep neural networks (Goodfellow et al., 2016), knowledge distillation (KD) (Ba & Caruana, 2014; Hinton et al., 2015) refers to the technique that uses well-trained networks to guide the training of another network. Typically, the well-trained network is named as the teacher network while the network to be trained is named as the student network. For distillation, the predictions from the teacher network are leveraged and referred to as the soft labels (Balan et al., 2015; Müller et al., 2019). Soft labels generated by the teacher network have been proven effective in large-scale empirical studies (Liang et al., 2019; Tian et al., 2020; Zagoruyko & Komodakis, 2017; Romero et al., 2015) as well as recent theoretical studies (Phuong & Lampert, 2019).

However, the reason why soft labels are beneficial to the student network is still not well explained. Giving a clear theoretical explanation is challenging: The optimization details of a deep network with the common one-hot labels are still not well-studied (Nagarajan & Kolter, 2019), not to mention training with the soft labels. Nevertheless, two recent studies (Müller et al., 2019; Yuan et al., 2020) shed light on the intuitions about how the soft labels work. Specifically, label smoothing, which is a special case of soft labels based training, is shown to regularize the activations of the penultimate layer to the network (Müller et al., 2019). The regularization property of soft labels is further explored in (Yuan et al., 2020). They hypothesize that in KD, one main reason why the soft labels work is the regularization introduced by soft labels. Based on the assumption, the authors

---

*These authors contributed equally to this work.

†Work done while the author was a research intern at Horizon Robotics.

design a teacher-free distillation method by turning the predictions of the student network into soft labels.

Considering that soft labels are targets for distillation, the evidence of the regularization brought by soft labels drives us to rethink soft labels for KD: Soft labels are both supervisory signals and regularizers. Meanwhile, it is known that there is a tradeoff between fitting the data and imposing regularizations, i.e., the bias-variance dilemma (Kohavi & Wolpert, 1996; Bishop, 2006), but it is unclear how bias and variance change for distillation with soft labels. Since the bias-variance tradeoff is an important issue in statistical learning, we investigate whether the bias-variance tradeoff exists for soft labels and how the tradeoff affects distillation performance.

We first compare the bias and variance decomposition of direct training with that of distillation with soft labels, noticing that distillation results in a larger bias error and a smaller variance. Then, we rewrite distillation loss into the form of a regularization loss adding the direct training loss. Through inspecting the gradients of the two terms during training, we notice that for soft labels, the bias-variance tradeoff varies sample-wisely. Moreover, by looking into a conclusion from (Müller et al., 2019), we observe that under the same temperature setting, the distillation performance is negatively associated with the number of some certain samples. These samples lead to bias increase and variance decrease and we name them as regularization samples. To investigate how regularization samples affect distillation, we first examine if we can design ad hoc filters for soft labels to avoid training with regularization samples. But completely filtering out regularization samples also deteriorates distillation performance, leading us to speculate that regularization samples are not well handled by standard KD. In the light of these findings, we propose weighted soft labels for distillation to handle the sample-wise bias-variance tradeoff, by adaptively assigning a lower weight to regularization samples and a larger weight to the others. To sum up, our contributions are:

- For knowledge distillation, we analyze how the soft labels work from a perspective of bias-variance tradeoff.
- We discover that the bias-variance tradeoff varies sample-wisely. Also, we discover that if we fix the distillation temperature, the number of regularization samples is negatively associated with the distillation performance.
- We design straightforward schemes to alleviate negative impacts from regularization samples and then propose the novel weighted soft labels for distillation. Experiments on large scale datasets validate the effectiveness of the proposed weighted soft labels.

## 2 RELATED WORKS

**Knowledge distillation.** Hinton et al. (2015) proposed to distill outputs from large and cumbersome models into smaller and faster models, which is named as knowledge distillation. The outputs for large networks are averaged and formulated as soft labels. Also, other kinds of soft labels have been widely used for training deep neural networks (Szegedy et al., 2016; Pereyra et al., 2017). Treating soft labels as regularizers were pointed out in (Hinton et al., 2015) since a lot of helpful information can be carried in soft labels. More recently, Müller et al. (2019) showed the adverse effect of label smoothing upon distillation. It is a thought-provoking discovery for the reason that both label smoothing and distillation are exploiting the regularization property behind soft labels. Yuan et al. (2020) further investigated the regularization property of soft labels and then proposed a teacher free distillation scheme.

**Distillation loss.** One of our main contributions is that we improve the distillation loss. For adaptively adjusting the distillation loss, Tang et al. (2019) pays attention to hard-to-learn and hard-to-mimic samples, and the latter is weighted based on the prediction gap between teacher and student. However, it does not consider that the teacher may give an incorrect guide to the student, under which the prediction gap is still large and such a method may lead to the performance being hurt. Saputra et al. (2019) transfers teacher's guidance only on the samples where the performance of the teacher surpasses the student, while Wen et al. (2019) deals with the incorrect guidance by probability shifting strategy. Our approach is different from the above methods, in terms of motivations as well as the proposed solutions.

**Bias-variance tradeoff.** Bias-variance tradeoff is a well-studied topic in machine learning (Kohavi & Wolpert, 1996; Domingos, 2000; Valentini & Dietterich, 2004; Bishop, 2006) and for neural

networks (Geman et al., 1992; Neal et al., 2018; Belkin et al., 2019; Yang et al., 2020). Existing methods are mainly concerned with the variance brought by the choice of network models. Our perspective is different from the previous methods since we focus on the behavior of samples during training. In our work, based on the results from Heskes (1998), we present the decomposition of distillation loss, which is defined by Kullback-Leibler divergence. Besides, our main contribution is not to study how to theoretically analyze the tradeoff, but how to adaptively tune the sample-wise tradeoff during training.

## 3 BIAS-VARIANCE TRADEOFF FOR SOFT LABELS

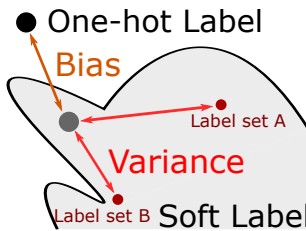

Figure 1: Bias and variance.

Soft labels play the role of supervisory signals and regularizations at the same time, which inspires us to rethink soft labels from the perspective of the bias-variance tradeoff. We begin our analysis with some mathematical descriptions. For a sample $\mathbf{x}$ labeled as $i$-th class, let the ground-truth label be a one-hot vector $\mathbf{y}$ where $y_i = 1$ and other entries are 0. Then for $x$ and softmax output temperature $\tau$, the soft label predicted by the teacher network is denoted as $\hat{y}_\tau^t$ and the output from the student is denoted as $\hat{y}_\tau^s$. The soft label $\hat{y}_\tau^t$ is then used for training the student by the distillation loss, i.e. $L_{\mathrm{kd}} = -\tau^2 \sum_k \hat{y}_{k,\tau}^t \log \hat{y}_{k,\tau}^s$, where $\hat{y}_{k,\tau}^s, \hat{y}_{k,\tau}^t$ means the $k$-th element of the student's output $\hat{y}_\tau^s$ and the teacher's output $\hat{y}_\tau^t$, respectively. With the above notations, the cross-entropy loss for training with one-hot labels is $L_{\mathrm{ce}} = -y_k \log \hat{y}_{k,1}^s$.

We now present the bias-variance decomposition for $L_{\mathrm{ce}}$ and $L_{\mathrm{kd}}$, based on the definition and notations from Heskes (1998). First, we denote the train dataset as $\mathcal{D}$ and the output distribution on a sample $\mathbf{x}$ of the network trained without distillation as $\hat{\mathbf{y}}_{\mathrm{ce}} = f_{\mathrm{ce}}(\mathbf{x}; \mathcal{D})$. For the network trained with distillation, the model also depends on the teacher network, so we define the output on $\mathbf{x}$ as $\hat{\mathbf{y}}_{\mathrm{kd}} = f_{\mathrm{kd}}(\mathbf{x}; \mathcal{D}, \mathcal{T})$, where $\mathcal{T}$ is the selected teacher network. Then, let the averaged output of $\hat{\mathbf{y}}_{\mathrm{kd}}$ and $\hat{\mathbf{y}}_{\mathrm{ce}}$ be $\bar{\mathbf{y}}_{\mathrm{kd}}$ and $\bar{\mathbf{y}}_{\mathrm{ce}}$, that is,

$$\bar{\mathbf{y}}_{\mathrm{ce}} = \frac{1}{Z_{\mathrm{ce}}} \exp(\mathbb{E}_{\mathcal{D}}[\log \hat{\mathbf{y}}_{\mathrm{ce}}]), \quad \bar{\mathbf{y}}_{\mathrm{kd}} = \frac{1}{Z_{\mathrm{kd}}} \exp(\mathbb{E}_{\mathcal{D}, \mathcal{T}}[\log \hat{\mathbf{y}}_{\mathrm{kd}}]), \tag{1}$$

where $Z_{\mathrm{ce}}, Z_{\mathrm{kd}}$ are two normalization constant. Then according to Heskes (1998), we have the following decomposition for the expected error on the sample $\mathbf{x}$ and $\mathbf{y} = t(\mathbf{x})$ is the ground truth label:

$$\begin{aligned} \mathrm{error}_{\mathrm{ce}} = \mathbb{E}_{\mathbf{x}, \mathcal{D}}\left[-\mathbf{y} \log \hat{\mathbf{y}}_{\mathrm{ce}}\right] &= \mathbb{E}_{\mathbf{x}, \mathcal{D}}\left[-\mathbf{y} \log \mathbf{y} + \mathbf{y} \log \frac{\mathbf{y}}{\bar{\mathbf{y}}_{\mathrm{ce}}} + \mathbf{y} \log \frac{\bar{\mathbf{y}}_{\mathrm{ce}}}{\hat{\mathbf{y}}_{\mathrm{ce}}}\right] \\ &= \mathbb{E}_{\mathbf{x}}[-\mathbf{y} \log \mathbf{y}] + \mathbb{E}_{\mathbf{x}}\left[\mathbf{y} \log \frac{\mathbf{y}}{\bar{\mathbf{y}}_{\mathrm{ce}}}\right] + \mathbb{E}_{\mathcal{D}}\left[\mathbb{E}_{\mathbf{x}}\left[\mathbf{y} \log \frac{\bar{\mathbf{y}}_{\mathrm{ce}}}{\hat{\mathbf{y}}_{\mathrm{ce}}}\right]\right] \quad (2) \\ &= \mathbb{E}_{\mathbf{x}}[-\mathbf{y} \log \mathbf{y}] + D_{\mathrm{KL}}(\mathbf{y}, \bar{\mathbf{y}}_{\mathrm{ce}}) + \mathbb{E}_{\mathcal{D}}[D_{\mathrm{KL}}(\bar{\mathbf{y}}_{\mathrm{ce}}, \hat{\mathbf{y}}_{\mathrm{ce}})] \\ &= \text{intrinsic noise} + \text{bias} + \text{variance}, \end{aligned}$$

where $D_{\mathrm{KL}}$ is the Kullback-Leibler divergence. The derivation of the variance term is based on the facts that $\frac{\log \bar{\mathbf{y}}_{\mathrm{ce}}}{\mathbb{E}_{\mathcal{D}}[\log \hat{\mathbf{y}}_{\mathrm{ce}}]}$ is a constant and $\mathbb{E}_{\mathbf{x}}[\mathbf{y}] = \mathbb{E}_{\mathbf{x}}[\bar{\mathbf{y}}_{\mathrm{ce}}] = 1$. Detailed derivations can be found from Eq. (4) in Heskes (1998). Next, we analyze the bias-variance decomposition of $L_{\mathrm{kd}}$. As mentioned above, when training with soft labels, extra randomness is introduced for the selection of a teacher network. In Fig. 1, we illustrate the corresponding bias and variance for the selection process of a set of soft labels, which are generated by a teacher network. In this case, a high variance model indicates the model (grey point) is closer to the one-hot trained model (black point), while a low variance model indicates that the model is closer to other possible models trained with soft labels (red points). Although for KD there are more sources introducing randomness, the overall variance brought by $L_{\mathrm{kd}}$ is not necessarily higher than $L_{\mathrm{ce}}$. In fact, existing empirical results strongly suggest that the overall variance is smaller with KD. For example, students trained with soft labels are better calibrated than one-hot baselines (Müller et al., 2019) and KD makes the predictions of students more consistent when facing adversarial noise (Papernot et al., 2016). Here, we present these empirical evidence as an assumption:

**Assumption 1** *The variance brought by KD is smaller than direct training, that is,* $\mathbb{E}_{\mathcal{D},\mathcal{T}}[D_{\mathrm{KL}}(\bar{\mathbf{y}}_{\mathrm{kd}}, \hat{\mathbf{y}}_{\mathrm{kd}})] \leqslant \mathbb{E}_{\mathcal{D}}[D_{\mathrm{KL}}(\bar{\mathbf{y}}_{\mathrm{ce}}, \hat{\mathbf{y}}_{\mathrm{ce}})].$

Similar to Eq. (2), we write the decomposition for $L_{\mathrm{kd}}$ as

$$\mathrm{error}_{\mathrm{kd}} = \mathbb{E}_{\mathbf{x}}[-\mathbf{y}\log\mathbf{y}] + D_{\mathrm{KL}}(\mathbf{y}, \bar{\mathbf{y}}_{\mathrm{ce}}) + \mathbb{E}_{\mathbf{x}}\left[\mathbf{y}\log\left(\frac{\bar{\mathbf{y}}_{\mathrm{ce}}}{\bar{\mathbf{y}}_{\mathrm{kd}}}\right)\right] + \mathbb{E}_{\mathcal{D},\mathcal{T}}[D_{\mathrm{KL}}(\bar{\mathbf{y}}_{\mathrm{kd}}, \hat{\mathbf{y}}_{\mathrm{kd}})]. \quad (3)$$

An observation here is that $\bar{\mathbf{y}}_{\mathrm{ce}}$ converges to one-hot labels while $\bar{\mathbf{y}}_{\mathrm{kd}}$ converges to soft labels, so $\bar{\mathbf{y}}_{\mathrm{ce}}$ is closer to the one-hot ground-truth distribution $\mathbf{y}$ than $\bar{\mathbf{y}}_{\mathrm{kd}}$, i.e., $\mathbb{E}_{\mathbf{x}}\left[\mathbf{y}\log\left(\frac{\bar{\mathbf{y}}_{\mathrm{ce}}}{\bar{\mathbf{y}}_{\mathrm{kd}}}\right)\right] \geqslant 0$. If we rewrite $L_{\mathrm{kd}}$ as $L_{\mathrm{kd}} = L_{\mathrm{kd}} - L_{\mathrm{ce}} + L_{\mathrm{ce}}$, then $L_{\mathrm{kd}} - L_{\mathrm{ce}}$ causes that the bias increases by $\mathbb{E}_{\mathbf{x}}\left[\mathbf{y}\log\left(\frac{\bar{\mathbf{y}}_{\mathrm{ce}}}{\bar{\mathbf{y}}_{\mathrm{kd}}}\right)\right]$ and the variance decreases by $\mathbb{E}_{\mathcal{D}}[D_{\mathrm{KL}}(\bar{\mathbf{y}}_{\mathrm{ce}}, \hat{\mathbf{y}}_{\mathrm{ce}})] - \mathbb{E}_{\mathcal{D},\mathcal{T}}[D_{\mathrm{KL}}(\bar{\mathbf{y}}_{\mathrm{kd}}, \hat{\mathbf{y}}_{\mathrm{kd}})]$.

From the above analysis, we separate $L_{\mathrm{kd}}$ into two terms, and $L_{\mathrm{kd}} - L_{\mathrm{ce}}$ leads to variance reduction, and $L_{\mathrm{ce}}$ leads to bias reduction. In the following sections, we first analyze how $L_{\mathrm{kd}} - L_{\mathrm{ce}}$ links to the bias-variance tradeoff during training. Then we analyze the changes in the relative importance between bias reduction and variance reduction during training with soft labels.

## 3.1 THE BIAS-VARIANCE TRADEOFF DURING TRAINING

It is known that bias reduction and variance reduction are often in conflict and we cannot minimize bias and variance together. However, if we consider the change of bias and variance during the training process, the importance of tuning the tradeoff also changes during training. Specifically, shortly after the training of the network starts, the bias error dominates the total error and the variance is less important. As training goes on, gradients of reducing the bias error (induced by $L_{\mathrm{ce}}$) and reducing the variance (induced by $L_{\mathrm{kd}} - L_{\mathrm{ce}}$) can be of the same scale for some samples, then we need to balance the tradeoff because reducing one term is likely to increase another one. Therefore for soft labels, we need to handle the bias-variance tradeoff in a sample-wise manner and take the training process into consideration.

To study the bias-variance tradeoff during training, we consider the gradients of bias and variance reduction. Let $\mathbf{z}$ be the logits output of the student on input $x$ and $z_i$ is $i$-th element of it, then we are interested in $\frac{\partial(L_{\mathrm{kd}} - L_{\mathrm{ce}})}{\partial z_i}$. For simplifying analysis, we are concerned with the gradients on the ground-truth related logit, that is, the sample $x$ is labeled as $i$-th class. Mathematically, for the gradients of variance reduction, we have

$$\frac{\partial(L_{\mathrm{kd}} - L_{\mathrm{ce}})}{\partial z_i} = \tau(\hat{y}_{i,\tau}^s - \hat{y}_{i,\tau}^t) - (\hat{y}_{i,1}^s - y_i) = \tau\left(\frac{e^{z_i/\tau}}{\sum_k e^{z_k/\tau}} - \hat{y}_{i,\tau}^t\right) - \left(\frac{e^{z_i}}{\sum_k e^{z_k}} - y_i\right), \quad (4)$$

where $\hat{y}_{i,\tau}^t$ denotes the $i$-th element of the teacher's prediction, i.e., $\hat{y}_\tau^t$. The term $L_{\mathrm{kd}} - L_{\mathrm{ce}}$ is easy to understand when $\tau = 1$ since the gradient now becomes $y_i - \hat{y}_{i,1}^t$. Meanwhile, for the bias reduction, we have $\frac{\partial L_{\mathrm{ce}}}{\partial z_i} = \hat{y}_{i,1}^s - y_i$, so $\frac{\partial L_{\mathrm{ce}}}{\partial z_i}$ and $\frac{\partial(L_{\mathrm{kd}} - L_{\mathrm{ce}})}{\partial z_i}$ always have different signs, leading to a tradeoff.

If $\frac{\partial L_{\mathrm{ce}}}{\partial z_i}$ is much higher than $\frac{\partial(L_{\mathrm{kd}} - L_{\mathrm{ce}})}{\partial z_i}$, the bias reduction dominates the overall optimization direction. Instead, if $\frac{\partial(L_{\mathrm{kd}} - L_{\mathrm{ce}})}{\partial z_i}$ becomes higher, the sample is used for variance reduction. Interestingly, we discover that under a fixed distillation temperature, the final performance is worse when more training samples are used for variance reduction, which will be introduced in the next section.

## 3.2 REGULARIZATION SAMPLES

Our analysis starts with a conclusion from Müller et al. (2019): *if a teacher network is trained with label smoothing, knowledge distillation into a student network is much less effective.* Inspired by the phenomenon, we gather the impact of bias and variance during training with different distillation settings. Let $a = \frac{\partial L_{\mathrm{ce}}}{\partial z_i}$ and $b = \frac{\partial(L_{\mathrm{kd}} - L_{\mathrm{ce}})}{\partial z_i}$, then as introduced before, we use $a$ and $b$ to represent the impact of bias and variance, respectively. If we have $|b| > |a|$ for a sample, we name the sample as a **regularization sample** since the variance dominates the optimization direction. From the collected data, we find that the number of regularization samples is closely related to distillation performance.

Table 1: We count the number of regularization samples with different distillation settings on CIFAR-100. The teacher-student network pair is WRN-40-2 (Zagoruyko & Komodakis, 2017) and WRN-16-2. Results are averaged over 5 repeated runs. The temperature column means the temperature for distillation and the label smoothing column means whether the teacher network is trained with label smoothing trick.

| Baseline Top-1 Acc | Teacher: 76.55 w/ label smoothing, 75.61 w/o label smoothing Student: 73.26 | | |
|---|---|---|---|
| Temperature | Label smoothing? | Student Top-1 Acc | Number of regularization samples |
| $\tau = 2$ | ✗ | **74.79** | 15379 |
| | ✓ | 74.62 | 25235 |
| $\tau = 4$ | ✗ | **74.92** | 17709 |
| | ✓ | 74.59 | 24775 |
| $\tau = 6$ | ✗ | **75.10** | 17408 |
| | ✓ | 74.46 | 24538 |

(a) Temperature $\tau = 2$     (b) Temperature $\tau = 4$     (c) Temperature $\tau = 6$

Figure 2: The number of regularization samples with respect to training epochs. The distillation settings are the same as the settings in Tab. 1.

In Tab. 1, we present the count of regularization samples for a student network trained by knowledge distillation. For distillation with a temperature higher than 1, which is the common setting, we observe that if the teacher network is trained with label smoothing, more samples will be involved in variance reduction. Also, distillation from a teacher trained with label smoothing performs worse, which is consistent with Müller et al. (2019). Therefore, we conclude that for distillation with soft labels, the regularization samples during training affect the final distillation performance.

Moreover, we plot the number of regularization samples with respect to different training epochs in Fig. 2. As demonstrated in the plots, the number of such samples increases much faster when using the teacher trained with label smoothing for distillation. For regularization samples, the gap of their number between with and without label smoothing becomes larger for more training epochs. These observations verify our motivation that the bias-variance tradeoff varies sample-wisely and evolves during the training process.

From the above results, we conclude that bias-variance tradeoff for soft labels varies sample-wisely, therefore the strategy for tuning the tradeoff should also be sample-wise. In the next section, we set up ad hoc filters for soft labels and further investigate how regularization samples affect distillation.

## 3.3 HOW REGULARIZATION SAMPLES AFFECT DISTILLATION

The results presented in the last section suggest that we should avoid training with regularization samples. Hence, we design two straightforward solutions and then find that totally filtering out regularization samples deteriorates the distillation performance.

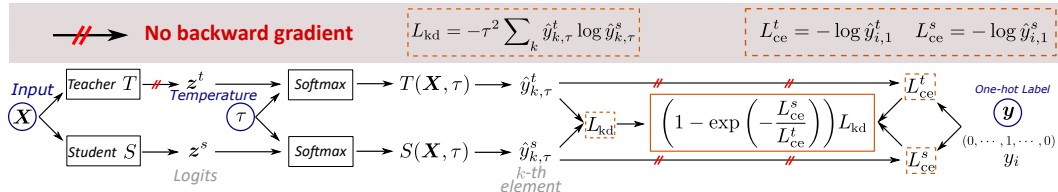

Figure 3: Computational graph of knowledge distillation with our proposed weighted soft labels.

Table 2: Study of the impact on distillation for regularization samples. Loss function presented here is for the loss on a specific sample. Results are classification Top-1 accuracy. We follow the settings used in Tab. 1 and set $\tau = 4$. Results are averaged over 5 runs.

| | | | Performance gap | |
|---|---|---|---|---|
| Setting | Loss function | Student performance | to direct training | to KD |
| | Teacher: 75.61; Student with direct training: 73.26; Student with standard KD: 74.92 | | | |
| Mask KD loss on the label related logit | $L_{\text{ce}} + L_{\text{kd}}^*$ | 73.51 | +0.25 | -1.41 |
| Excluding regularization samples | $\begin{cases} L_{\text{ce}}, \text{if } \|a\| < \|b\| \\ L_{\text{ce}} + L_{\text{kd}}, \text{if } \|a\| \geqslant \|b\| \end{cases}$ | 74.59 | +1.33 | -0.33 |
| Only on regularization samples | $\begin{cases} L_{\text{ce}}, \text{if } \|a\| \geqslant \|b\| \\ L_{\text{ce}} + L_{\text{kd}}, \text{if } \|a\| < \|b\| \end{cases}$ | 73.86 | +0.60 | -1.06 |

The first experiment we conduct is to manually resolve the conflicting gradient on the label related logit, as defined in section 3.2. Specifically, we apply a mask to the distillation loss $L_{\text{kd}}$ such that $\frac{\partial L_{\text{kd}}}{\partial z_i} = 0$ where $i$ is the label. Consequently, the loss for this sample now becomes $L_{\text{kd}}^* = \sum_{k \neq i} \hat{y}_{k,\tau}^t \log \hat{y}_{k,\tau}^s$. The motivation behind the masked distillation loss is that we only transfer the knowledge of resemblances among the labels. Another experiment is to figure out what role in distillation those regularization samples will play. To investigate this, we carry out knowledge distillation on two subsets of samples: 1) $L_{\text{kd}}$ is not valid on regularization samples, and 2) $L_{\text{kd}}$ is valid only on regularization samples.

The results of the two experiments are presented in Tab. 2. We can observe that all of the three approaches are not as good as the baseline knowledge distillation performance, but are better than the direct training baseline. First, since masking $L_{\text{kd}}$ loss on the label related logit results in worse performance compared to standard KD, we cannot resolve the tradeoff by applying a mask on the ground truth related logit. Then, from the second experiment, we can see that filtering out regularization samples deteriorates the distillation performance. Moreover, the result of the third experiment is higher than the direct training baseline, indicating that regularization samples are still valuable for distillation. The above results motivate us to think that regularization samples are not fully exploited by standard KD and we can tune the tradeoff to fulfill the potential of regularization samples.

## 4 WEIGHTED SOFT LABELS

From the last section, we realize that the bias-variance tradeoff varies sample-wisely during training and under fixed distillation settings, the number of regularization samples is negatively associated with the final distillation performance. Yet, discarding regularization samples deteriorates distillation performance and distilling knowledge from these samples is better than the direct training baseline. The above evidence inspires us to lower the weight of regularization samples.

Recall that regularization samples are defined by the relative value of $a$ and $b$, we propose to assign importance weight to a sample according to $a$ and $b$. However, since $L_{\text{kd}}$ is computed with the hyperparameter temperature, $a$ and $b$ are correlated with the temperature and thus bring difficulty to tuning the hyperparameter. To make the weighting scheme independent of the temperature hyperparameter, we compare $a$ and $b$ with temperature $\tau = 1$. Note that when $\tau = 1$, $a = \hat{y}_{i,1}^s - y_i$ and $b = y_i - \hat{y}_{i,1}^t$, so we compare $\hat{y}_{i,1}^s$ and $\hat{y}_{i,1}^t$ instead. Finally, in the light of previous works that assign sample-wise weights (Lin et al., 2017; Tang et al., 2019), we propose weighted soft labels for

Table 3: Top-1 classification accuracy results on CIFAR-100. Comparison results are quoted from Tian et al. (2020). We report our results over 5 repeated runs.

| | Same architecture style | | | | | Different architecture style | | |
|---|---|---|---|---|---|---|---|---|
| Teacher | WRN-40-2 | resnet56 | resnet110 | resnet110 | resnet32x4 | resnet32x4 | resnet32x4 | WRN-40-2 |
| Student | WRN-40-1 | resnet20 | resnet20 | resnet32 | resnet8x4 | ShuffleNetV1 | ShuffleNetV2 | ShuffleNetV1 |
| Teacher | 75.61 | 72.34 | 74.31 | 74.31 | 79.42 | 79.42 | 79.42 | 75.61 |
| Student | 71.98 | 69.06 | 69.06 | 71.14 | 72.50 | 70.5 | 71.82 | 70.5 |
| FitNet | 72.24 | 69.21 | 68.99 | 71.06 | 73.50 | 73.59 | 73.54 | 73.73 |
| AT | 72.77 | 70.55 | 70.22 | 72.31 | 73.44 | 71.73 | 72.73 | 73.32 |
| SP | 72.43 | 69.67 | 70.04 | 72.69 | 72.94 | 73.48 | 74.56 | 74.52 |
| CC | 72.21 | 69.63 | 69.48 | 71.48 | 72.97 | 71.14 | 71.29 | 71.38 |
| VID | 73.30 | 70.38 | 70.16 | 72.61 | 73.09 | 73.38 | 73.40 | 73.61 |
| RKD | 72.22 | 69.61 | 69.25 | 71.82 | 71.90 | 72.28 | 73.21 | 72.21 |
| PKT | 73.45 | 70.34 | 70.25 | 72.61 | 73.64 | 74.10 | 74.69 | 73.89 |
| AB | 72.38 | 69.47 | 69.53 | 70.98 | 73.17 | 73.55 | 74.31 | 73.34 |
| FT | 71.59 | 69.84 | 70.22 | 72.37 | 72.86 | 71.75 | 72.50 | 72.03 |
| FSP | n/a | 69.95 | 70.11 | 71.89 | 72.62 | n/a | n/a | n/a |
| NST | 72.24 | 69.60 | 69.53 | 71.96 | 73.30 | 74.12 | 74.68 | 74.89 |
| KD | 73.54 | 70.66 | 70.67 | 73.08 | 73.33 | 74.07 | 74.45 | 74.83 |
| CRD | 74.14 | 71.16 | 71.46 | 73.48 | 75.51 | 75.11 | 75.65 | 76.05 |
| *Ours* | **74.48** | **72.15** | **72.19** | **74.12** | **76.05** | **75.46** | **75.93** | **76.21** |

knowledge distillation, which is formally defined as

$$L_{\text{wsl}} = \left(1 - \exp\left(-\frac{\log \hat{y}_{i,1}^s}{\log \hat{y}_{i,1}^t}\right)\right) L_{\text{kd}} = \left(1 - \exp\left(-\frac{L_{\text{ce}}^s}{L_{\text{ce}}^t}\right)\right) L_{\text{kd}}, \tag{5}$$

where $i$ is the ground truth class of the sample. The above equation means that a weighting factor is assigned to each sample's $L_{\text{kd}}$ according to the predictions of the teacher and the student. In this way, if compared to the teacher, a student network is relatively better trained on a sample, we have $\hat{y}_{i,1}^s > \hat{y}_{i,1}^t$, then a smaller weight is assigned to this sample. In Fig. 3, the whole computational graph of knowledge distillation with the proposed weighted soft labels is demonstrated. Finally, we add $L_{\text{wsl}}$ and $L_{\text{ce}}$ together to supervise the network, i.e., $L_{\text{total}} = L_{\text{ce}} + \alpha L_{\text{wsl}}$, where $\alpha$ is a balancing hyperparameter.

## 5 EXPERIMENTS

To evaluate our weighted soft labels comprehensively, we first conduct experiments with various teacher-student pair settings on CIFAR-100 (Krizhevsky et al., 2009). Next, we compare our method with current state-of-the-art distillation methods on ImageNet (Deng et al., 2009). To validate the effectiveness of our method in terms of handling the bias-variance tradeoff, we conduct ablation experiments by applying weighted soft labels on different subsets.

### 5.1 DATASET AND HYPERPARAMETER SETTINGS

The datasets used in our experiments are CIFAR-100 (Krizhevsky et al., 2009) and ImageNet (Deng et al., 2009). CIFAR-100 contains 50K training and 10K test images of size $32 \times 32$. ImageNet contains 1.2 million training and 50K validation images. Except the loss function, training settings like learning rate or training epochs are the same with Tian et al. (2020) for CIFAR-100 and Heo et al. (2019) for ImageNet. For distillation, we set the temperature $\tau = 4$ for CIFAR and $\tau = 2$ for ImageNet. For loss function, we set $\alpha = 2.25$ for distillation on CIFAR and $\alpha = 2.5$ for ImageNet via grid search. The teacher network is well-trained previously and fixed during training.

For comparison, the following recent state-of-the-art methods are chosen: FitNet (Romero et al., 2015), AT (Zagoruyko & Komodakis, 2017), SP (Tung & Mori, 2019), CC (Peng et al., 2019), VID (Ahn et al., 2019), RKD (Park et al., 2019), PKT (Passalis & Tefas, 2018), AB (Heo et al., 2019),

Table 4: Top-1 and Top-5 classification accuracy results on ImageNet validation set. All training hyperparameter like learning rate and training epochs are in accordance with (Heo et al., 2019).

| Teacher: ResNet-34 → Student: ResNet-18 | | | Teacher: ResNet-50 → Student: MobileNet-v1 | | |
|---|---|---|---|---|---|
| Method | Top-1 Acc | Top-5 Acc | Method | Top-1 Acc | Top-5 Acc |
| Teacher | 73.31 | 91.42 | Teacher | 76.16 | 92.87 |
| Student | 69.75 | 89.07 | Student | 68.87 | 88.76 |
| KD | 70.67 | 90.04 | KD | 70.49 | 89.92 |
| AT | 71.03 | 90.04 | AT | 70.18 | 89.68 |
| NST | 70.29 | 89.53 | FT | 69.88 | 89.5 |
| FSP | 70.58 | 89.61 | AB | 68.89 | 88.71 |
| RKD | 70.40 | 89.78 | RKD | 68.50 | 88.32 |
| Overhaul | 71.03 | 90.15 | Overhaul | 71.33 | 90.33 |
| CRD | 71.17 | 90.13 | CRD | 69.07 | 88.94 |
| *Ours* | **72.04** | **90.70** | *Ours* | **71.52** | **90.34** |

FT (Kim et al., 2018), FSP (Yim et al., 2017), NST (Huang & Wang, 2017), Overhaul (Heo et al., 2019) and CRD (Tian et al., 2020).

## 5.2 MODEL COMPRESSION

**Results on CIFAR-100**   In Tab. 3, we present the Top-1 classification accuracy of our method and comparison methods. The results of comparison methods are quoted from Tian et al. (2020). Teacher-student pairs of the same and different architecture styles are considered. For pairs of same architecture style, we use wide residual networks (Zagoruyko & Komodakis, 2017) and residual networks (He et al., 2016). For pairs of different architecture style, residual networks and ShuffleNet (Zhang et al., 2018) pairs are chosen for experiments. As shown in the table, for distillation with both same and different architecture style, our method reached new state-of-the-art results. Specifically, our method outperforms standard KD by a large margin, which verifies the effectiveness of our method.

**Results on ImageNet**   In Tab. 4, we compare our method with current SOTA methods on ImageNet. Note that for the ResNet34 → ResNet-18 distillation setting, the result of CRD is trained 10 more extra epochs while ours is the same as other methods. For ResNet-50 → MobileNet-v1 distillation setting, NST and FSP are not chosen for comparison as the two methods require too large GPU memories, so we include the accuracy of FT and AB reported in Heo et al. (2019) for comparison. Our results outperform all the existing methods, verifying the practical value of our method.

## 5.3 ABLATION STUDIES

**Weighted soft labels on different subsets.**   Recall that we propose weighted soft labels for tuning sample-wise bias-variance tradeoff, it is still unclear whether the improvements come from a well-handled sample-wise bias-variance tradeoff. To investigate this issue, we compare the performance gain of weighted soft labels on different training subsets. Similar to the settings used in Tab. 2, we apply weighted soft labels on two different subsets: only the regularization samples and excluding regularization samples. In Tab. 5, we show

Table 5: Performance on different subsets with soft labels and our weighted soft labels. RS means regularization samples. Results are averaged over 5 runs.

| Subsets | Standard KD | Weighted |
|---|---|---|
| Only on RS | 73.86 | 74.46 |
| Excluding RS | 74.59 | 75.35 |

the results on subsets of only regularization samples and excluding regularization samples. From the significant improvements, we can see that our method can not only improve performance on the RS subset, the improvements on excluding RS subset is also significant. We conclude that weighted soft labels can tune sample-wise bias-variance tradeoff globally and lead to an improved distillation performance.

**Distillation with label smoothing trained teacher.** Our exploration of bias-variance tradeoff starts with the conclusion made in Müller et al. (2019): a teacher network trained with the label smoothing trick is less effective for distillation. It is worthwhile to study whether the conclusion remains true for distillation with our weighted soft labels. As discussed before, we hold the opinion that too many regularization samples make the distillation less effective. Since our weighted soft label is proposed to mitigate the negative effects of the regularization samples, with the same settings from Tab. 1, we conduct comparison experiments in Tab. 6 to see if the negative effects still exist.

Table 6: Distillation using weighted soft labels and teacher trained with label smoothing (denoted as *LS?*). Results are averaged over 5 runs.

| $\tau$ | *LS?* | Standard KD | Weighted |
|---|---|---|---|
| 4 | ✗ | 74.92 | 75.78 |
| 4 | ✓ | 74.59 | 75.60 |
| 6 | ✗ | 75.10 | 75.74 |
| 6 | ✓ | 74.46 | 75.42 |

It is evident that weighted soft labels significantly improve the distillation performance, especially for distillation from the teacher trained with label smoothing. Besides, using the teacher trained with label smoothing still performs worse than that without label smoothing, which again verifies the conclusion drawn by Müller et al. (2019).

## 6 CONCLUSION

Recent studies (Müller et al., 2019; Yuan et al., 2020) point out that one important reason behind the effectiveness of distillation is the regularization effect brought by being soft. In this paper, we rethink the soft labels for distillation from a bias-variance tradeoff perspective. The tradeoff varies sample-wisely and we propose weighted soft labels to handle the tradeoff, of which the effectiveness is verified with experiments on standard evaluation benchmarks.

## ACKNOWLEDGEMENTS

This work is supported in part by a gift grant from Horizon Robotics and National Science Foundation Grant CNS-1951952. We thank Yichen Gong, Chuan Tian, Jiemin Fang and Yuzhu Sun for the discussion and assistance.

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

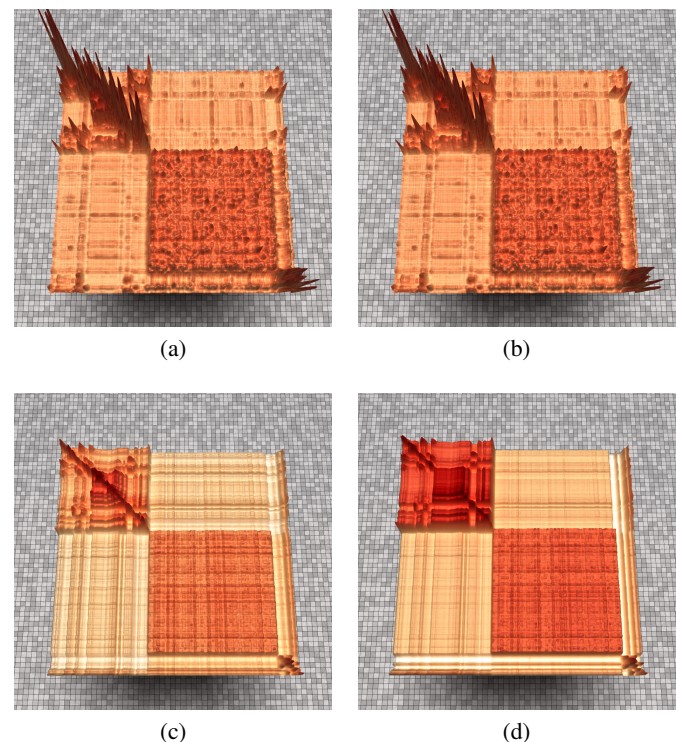

Figure 4: Visualization of the resemblances introduced by soft label regularizers: (a) VGG-19 (Teacher) $\rightarrow$ VGG-16 (Student), (b) ResNet-50 (Teacher) $\rightarrow$ ResNet-18 (Student). And semantic similarity between label names: (c) LCH similarity (Pedersen et al., 2004), (d) WUP similarity (Pedersen et al., 2004). Darker areas denote larger values.

Sergey Zagoruyko and Nikos Komodakis. Paying more attention to attention: Improving the performance of convolutional neural networks via attention transfer. In *International Conference on Learning Representations*, 2017.

Xiangyu Zhang, Xinyu Zhou, Mengxiao Lin, and Jian Sun. Shufflenet: An extremely efficient convolutional neural network for mobile devices. In *Proceedings of the IEEE conference on computer vision and pattern recognition*, pp. 6848–6856, 2018.

# A  APPENDIX

## A.1  VISUALIZATION OF THE RESEMBLANCES INTRODUCED BY SOFT LABEL REGULARIZER

In Sec 3.1, we propose to study $\frac{\partial(L_{\mathrm{kd}}-L_{\mathrm{ce}})}{\partial z_i}$ during the training process. When $\tau = 1$, we show that $\frac{\partial(L_{\mathrm{kd}}-L_{\mathrm{ce}})}{\partial z_i}$ equals $y_i - \hat{y}_{i,1}^t$. As $y_{i,\tau}^t$ is the output from the teacher network and computed by a linear mapping of the activations in the teacher's penultimate layer, the regularization indicates that the student should follow the learned the resemblances between classes (Hinton et al., 2015; Müller et al., 2019). Still, two questions are unclear: 1) what the resemblances are and 2) whether the regularization still indicates resemblances if $\tau$ is set to 4, a widely adopted hyperparameter (Tian et al., 2020).

Towards answering the questions, we visualize the value of gradient vector $\frac{\partial(L_{\mathrm{kd}}-L_{\mathrm{ce}})}{\partial z}$ concerning each class. Specifically, on ImageNet (Deng et al., 2009) training set and $\tau = 4$, we calculate the average value of $\frac{\partial(L_{\mathrm{kd}}-L_{\mathrm{ce}})}{\partial z}$ for each class. Let $M$ be the matrix of values with the $ij$-th entry $M_{ij}$ means averaged $\frac{\partial(L_{\mathrm{kd}}-L_{\mathrm{ce}})}{\partial z_i}$ for class $j$. Since $\sum_i M_{ij} = 0$, diagonal elements are ignored for visualization. The results are visualized in Fig. 4. We find that plotting the common correlation

Table 7: Intermediate states between excluding and only on regularization samples

(a) CIFAR100 (WRN-40-2→WRN-40-1 with KD)

| Percentage of excluded regularization samples. | | | |
|---|---|---|---|
| 100% | 75% | 50% | 25% |
| 74.59 | 74.63 | 74.72 | 74.87 |

| Percentage of adding non-regularization samples | | | |
|---|---|---|---|
| 0% | 25% | 50% | 75% |
| 73.86 | 74.12 | 74.47 | 74.71 |

(b) CIFAR100 (WRN-40-2→WRN-40-1 with weighted soft labels)

| Percentage of excluded regularization samples | | | |
|---|---|---|---|
| 100% | 75% | 50% | 25% |
| 75.35 | 75.48 | 75.61 | 75.72 |

| Percentage of adding non-regularization samples | | | |
|---|---|---|---|
| 0% | 25% | 50% | 75% |
| 74.46 | 74.79 | 75.18 | 75.53 |

matrix heatmap is ambiguous, because the matrix to be visualized is of large size ($1000 \times 1000$) with a large variance. By treating each entry $M_{ij}$ as a vertex and then constructing a mesh for the matrix, we apply subdivision (Loop, 1987) to the mesh for smoothing the extreme points and finally rendering the mesh by ray-tracing package PlotOptiX. We can observe the several facts from the figures: 1) Comparing the sub-figure (a) and (b), we can see that for distillation resemblances implied by regularizers are similar across different teacher-student pairs. 2) Comparing (ab) with (cd), we can see that the resemblances are consistent with the semantic similarity of image class names.

In a word, for $\tau = 4$, the variance reduction brought by soft labels still implies resemblances among labels, which are consistent with the semantic distance of class names. In the next section, we will analyze how bias-variance tradeoff changes when training with soft labels.

## A.2 INTERMEDIATE STATES BETWEEN EXCLUDING AND ONLY ON REGULARIZATION SAMPLES

To further investigate the phenomenon about regularization samples, we conduct experiments to show the intermediate states between excluding and only on regularization samples. Two settings are considered here: First, we gradually exclude regularization samples during training, from excluding all regularization samples to excluding 25% regularization samples; Second, we keep all regularization samples and then gradually add non-regularization samples. Since we judge a sample is regularization or not according to the training loss, we cannot pre-define a sample set such that a certain percentage samples are kept or dropped. Therefore, we propose to conduct these experiments by assigning a probability to whether backward the loss computed with regularization samples. For example, if during training, a sample is marked as regularization sample according to the value of $a$ and $b$, we backward the loss of this sample by a probability $p = 0.5$. In this way, we can get the performance of excluding 75% regularization samples. In Tab. 7, we first present result with KD in (a) and then present result with weighted soft labels applied in (b). We can observe that weighted soft labels are indeed balancing the sample-wise, not on dataset scale, bias and variance.

## A.3 COMBINING WITH RKD (PARK ET AL., 2019).

To investigate how the weighted soft labels can be applied to the variants of KD, we conduct an experiment of combining RKD (Park et al., 2019) with our weighted soft labels. Relational knowledge distillation measures the L2 distance of features between two samples or the angle formed by three samples as knowledge to transfer. In other words, the knowledge in RKD is measured by

Table 8: Combining weighted soft labels with RKD (Park et al., 2019).

| Distillation settings | WRN-40-2 → WRN-16-2 | WRN-40-2 → WRN-40-1 | resnet56 → resnet20 |
|---|---|---|---|
| Teacher | 75.61 | 75.61 | 72.34 |
| Student | 73.26 | 71.98 | 69.06 |
| RDK | 74.12 | 73.34 | 70.25 |
| WSL + RKD | 74.65 | 73.89 | 70.73 |

Table 9: Comparison to other weighting forms. (Setting: CIFAR100, WRN-40-2→WRN-40-1)

| $\alpha$ | 2.0 | 3.0 | 4.0 |
|---|---|---|---|
| Sigmoid baseline | 74.13 | 73.97 | 73.29 |
| Ours | 74.38 | 74.12 | 73.46 |

the relations between sample pairs. It is no longer sample-independent, which is different from the weighted soft labels applied to KD which can assign the weights sample-wisely. We currently take the averaged weighting factors of the involved sample pairs when calculating the distance/angle matrix. The results on CIFAR-100 are presented in Tab. 8 (averaged over 5 runs). As can be observed from the table, the weighted soft label applied to RKD still brings improvements, though not that big compared with WSL applied to KD. Also, we believe that it is an important future direction to explore the applications to more variants of KD.

### A.4    OTHER VARIANTS OF WEIGHTING.

In the work, the weighting scheme is defined as $\left(1 - \exp\left(-\frac{L_{ce}^s}{L_{ce}^t}\right)\right)$, which is inspired by Lin et al. (2017); Tang et al. (2019). The basic idea is to convert $\frac{L_{ce}^s}{L_{ce}^t}$ into a value in $[0, 1]$, so that the weights of regularization samples are lower than those non-regularization samples. A straightforward baseline is that we can use the Sigmoid function to convert $\frac{L_{ce}^s}{L_{ce}^t}$ into a value in $[0, 1]$. Note that $\frac{L_{ce}^s}{L_{ce}^t}$ is always bigger than 1, so the weight needs scaling and can be defined as $\frac{2}{1+\exp(-\frac{L_{ce}^s}{L_{ce}^t})} - 1$. In Tab. 9, we present the comparison between adopted weighting form and the Sigmoid baseline. We can see that as long as we can adaptively tune the sample-wise bias-variance tradeoff, the performance is better than KD, i.e., without weighted soft labels. Therefore, although the proposed weighting form is not mathematically optimal, the not-too-big or not-too-small weights for these regularization examples are not hard to tune. These results verify our main contribution that there is sample-wise bias-variance tradeoff and we need to assign weights to the regularization examples.

### A.5    ABLATION ON $\alpha$.

In Tab. 10 We first tune the value of $\alpha$ on CIFAR100, with four values $\{1, 2, 3, 4\}$ tested. Then we test with three values in $[2, 3]$ in (b). Finally, we tune $\alpha$ on ImageNet in (c). As a conclusion, the results are not very sensitive to $\alpha$ and the cost of searching $\alpha$ in our work is not expensive.

### A.6    RESULTS ON MULTINLI

To further validate our method, we conduct experiments on an NLP dataset MultiNLI (Williams et al., 2018). In this setting, the teacher is BERT-base-cased with 12 layers, 768 Hidden and 108M params. The student is T3 with 3 layers, 768 Hidden and 44M params. Besides, we follow the training setting in Sun et al. (2019). In Tab. 11, we present the result comparisons of standard KD and our weighted soft labels.

Table 10: Ablation on $\alpha$.

(a) CIFAR100 (WRN-40-2$\rightarrow$WRN-40-1)

| $\alpha$ | 1 | 2 | 3 | 4 |
|---|---|---|---|---|
| Top1 | 73.67 | 74.38 | 74.12 | 73.46 |

(b) CIFAR100 (WRN-40-2$\rightarrow$WRN-40-1)

| $\alpha$ | 2.25 | 2.5 | 2.75 |
|---|---|---|---|
| Top1 | 74.48 | 74.34 | 74.21 |

(c) ImageNet (ResNet-34$\rightarrow$ResNet-18)

| $\alpha$ | 2 | 2.25 | 2.5 |
|---|---|---|---|
| Top1 | 71.91 | 71.96 | 72.04 |

Table 11: Results on MultiNLI.

| | Teacher (BERT-12) | Student (BERT-3) | KD (BERT-3) | Ours (BERT-3) |
|---|---|---|---|---|
| Results reported by Sun et al. (2019) | 83.7 | 74.8 | 75.4 | - |
| Our replications | 83.57 | 75.06 | 75.50 | 76.28 |

