# OpenReview forum: "Rethinking Soft Labels for Knowledge Distillation: A Bias–Variance Tradeoff Perspective"
_ICLR.cc/2021/Conference — ICLR 2021 Poster_

### Official Review · AnonReviewer2 · 2020-10-20

**Rating:** 6
**Confidence:** 4

**Review:**

Summary:

This paper analyzes the distillation from the bias-variance perspective. Beyond this, the regularization samples affect the performance. Based on the observation, a novel weighted mechanism is proposed to distill knowledge from teacher networks.

Strengths:

+) This paper is clear and easy to follow, the organization is good. The bias-variance analysis, the regularization samples, the weighted soft labels all make sense. I felt comfortable when I was reading this paper.

+) The analysis is clear and reasonable. The deduction seems correct. The figures are clear.

+) The experiments are enough to examine the effectiveness of the proposed weighted distillation (see below).

+) The code is submitted to contribute to the community. I appreciate the submission.


Weaknesses & Concerns:

-) Sec. 3, first paragraph, $T(x, \tau)$ -> $\hat{y}^t = T(x, \tau)$, $S(x, \tau)$ -> $\hat{y}^s = S(x, \tau)$ to make it more clear.

-) ' For loss function, we set α = 2.25 for distillation on CIFAR and α = 2.5 for ImageNet via grid search.' How many $\alpha$s have been tested? What are the results? The main concern is that the grid search is costly in practice.  Therefore, I appreciate the analysis in this paper that helps us understanding KD better. However, the grid searched hyper-parameters makes Sec. 4 costly in practice.

Based on the quality of the paper, I select 6 as the initial score.

---

> ### Author Response · Authors · 2020-11-14
> **Author response to AnonReviewer2**
>
> We thank the reviewer for the positive feedback and constructive comments.
>
> ---
> - Q1: Sec. 3, make notations more clear.
> - A1: Thanks for the suggestion. We revised the first paragraph and other parts to avoid the abuse of notations (as R3 suggested). Also, we use $\hat{y}^s, \hat{y}^t$ instead of $S(x, \tau),T(x, \tau)$ to make notation consistent.
>
> ---
> - Q2: 'For loss function, we set alpha=2.25 for distillation on CIFAR and alpha=2.5 for ImageNet via grid search.' How many $\alpha$ have been tested? What are the results? The main concern is that the grid search is costly in practice. Therefore, I appreciate the analysis in this paper that helps us understanding KD better. However, the grid searched hyper-parameters makes Sec. 4 costly in practice.
> - A2: Thanks for the comments. The searched hyper-parameters are not costly in practice. First, note that the weight $\left(1-\exp\left(-\frac{L_{ce}^s}{L_{ce}^t}\right)\right)$ is always smaller than 1. And $\alpha$ is used for balancing cross-entropy and distillation, so $alpha$ should be larger than 1 and we first search the hyper-parameters on CIFAR100 by setting $\alpha=\{1,2,3,4\}$, results are ((WRN-40-2$\rightarrow$WRN-40-1))
>
> |$\alpha$|1|2|3|4|
> |---|---|---|---|---|
> |Top1|73.67|74.38|74.12|73.46|
>
> Then test $\alpha$ from 2 to 3, we get
>
> |$\alpha$| 2.25| 2.5| 2.75|
> |---|---|---|---|
> |Top1|74.48|74.34|74.21|
>
> Thus we set $\alpha=2.25$ for CIFAR100. For ImageNet, we run 3 experiments around 2.25 (ResNet-34$\rightarrow$ResNet-18)
>
> |$\alpha$| 2|2.25|2.5|
> |---|---|---|---|
> |Top1|71.91|71.96|72.04|
>
> So we use $\alpha=2.5$ for ImageNet. Overall, we find that the impact of $\alpha$ is not significant within range 0.5.
> In conclusion, the results are not very sensitive to $\alpha$ and the cost of searching in our work is not expensive. These results are updated in the appendix section.

---

### Official Review · AnonReviewer4 · 2020-10-23
**Official Blind Review #4**

**Rating:** 7
**Confidence:** 3

**Review:**

In this paper, the authors studied the soft labels for knowledge distillation from a bias-variance tradeoff perspective. Specifically, the authors first provide a mathematically descriptions of the bias-variance decomposition in knowledge distillation. Then, based on the theoretically analysis and experiments, the authors proposed an novel weighted soft labels to help the network adaptively handle the sample-wise bias-variance tradeoff.

Strength:
1. The paper is well written and easy to follow.

2. The authors provide many mathematically proof in the paper, which could serve as a theoretically foundation of this topic.

3. The authors claim that their code is available in the supplementary, which makes it easier for other researchers to reproduce this work.

Weakness:
1. In addition to computer vision, Knowledge distillation is also very popular in NLP area recently. Maybe the authors can add more experiments for NLP to prove that the solution can be widely-adopted.

2. I am not very familiar with computer vision area, but it seems that ResNet-50/ResNet-34 has a huge gap with the current SOTA (Ref: https://paperswithcode.com/sota/image-classification-on-imagenet). Experimenting with stronger baselines would make the conclusion even stronger.

Overall comments:
I think this is a good paper and I'd like to see it to be accepted.

---

> ### Author Response · Authors · 2020-11-14
> **Author response to AnonReviewer4**
>
> We thank the reviewer for the helpful feedback.
>
> ---
> - Q1: In addition to computer vision, Knowledge distillation is also very popular in NLP area recently. Maybe the authors can add more experiments for NLP to prove that the solution can be widely-adopted.
> - A1: Thanks for this suggestion. We are currently implementing our method with an NLP distillation framework [1]. The setting is as follows:
>
> dataset:  multinli (MNLI);
> teacher: BERT-base-cased, 12 layers, 768 Hidden, 108M params;
> student: T3, 3 layers, 768 Hidden, 44M params;
>
> Besides, we follow the training setting in [2]. The results reported in [2]:
>
> |teacher-BERT-12|baseline-BERT-3|KD-BERT-3|
> |---|---|---|
> |83.7|74.8|75.4|
>
> Our replication and our results:
>
> |teacher-BERT-12|baseline-BERT-3|KD-BERT-3|**Ours-BERT-3**|
> |---|---|---|---|
> |83.57|75.06|75.50|76.28|
>
> Also, we update our code for reproducing NLP results. Please check our code for detailed training settings.
>
> ---
> - Q2: I am not very familiar with computer vision area, but it seems that ResNet-50/ResNet-34 has a huge gap with the current SOTA (Ref: https://paperswithcode.com/sota/image-classification-on-imagenet). Experimenting with stronger baselines would make the conclusion even stronger.
> - A2: In the experiments, we run ResNet-50/ResNet-34 to have a fair comparison with related works. In fact, we also tried our method on stronger baselines [2], but the results are not listed in the paper due to the lack of comparison distillation methods.
>
> |Student| 70.93|Teacher|74.40|
> |---|---|---|---|
> |Method|Teacher|Student|	top1 acc|
> |KD	|res34	|res18|	71.86|
> |AT	|res34	|res18|	71.97|
> |Ours|res34	|res18|	72.83|
>
> These results demonstrate that our method is also scalable to those better baselines, such as those models suggested by the link provided by the reviewer.
>
> [1] https://github.com/airaria/TextBrewer
>
> [2] Sun, Siqi, et al. "Patient Knowledge Distillation for BERT Model Compression."
>
> [3] https://cv.gluon.ai/model_zoo/classification.html

---

### Official Review · AnonReviewer3 · 2020-10-28
**a new perspective of performing knowledge distillation using bias-variance decomposition**

**Rating:** 4
**Confidence:** 3

**Review:**

The paper shows a new perspective of tackling the knowledge distillation problem. The author(s) have decomposed the expected student's training error into the bias, variance, and irreducible noise parts. This decomposition is further rewritten as two parts: one for bias reduction and another for variance reduction. The motivation is clearly explained and the experimental results show that this new approach can improve the model training performance of the student on both CIFAR100 and Imagenet.

pros:
- A novel perspective of performing knowledge distillation.
- The motivation is clear and the framework is straightforward to understand.
- The code is provided, therefore, I believe the reproducibility is high.

cons:
- The definition of variance is flawed. The formulation doesn't match with the figure illustration and code implementation.
- The observed phenomenon regarding the regularization examples is not well explained.
- The experimental details are not sufficient.

concerns:
- My major concern comes from the definition of variance and its corresponding implementation. From my understanding, the variance should be estimated from multiple teachers, just as shown in Fig 1, which matches my thoughts. But it seems like the author(s) are trying to fade this concept intentionally. First, the expectation's definition for Equation 1 is not clear. It should take the expectation over the space of infinite teachers, or at least empirically multiple teachers with a substantial number. Second, after checking on the code, I only find one teacher is utilized in the implementation. Would you please elaborate on why this is the design?

- Second concern is that the mathematical reasons behind the not-too-big or not-too-small weights for these regularization examples are unclear. This reweight scheme is purely empirically, and it would be a much solid paper with mathematical interpretations.

- The bias-variance decomposition on classification has been well studied in multiple papers, e.g., [Pedro Domingos 2002], and [Giorgio Valentini 2004]. The relation (similarity and difference) between your employed BVD from [Heskes 1998] and [Pedro Domingos 2002] is not discussed. The latter [Giorgio Valentini 2004] is not even cited, which seems inappropriate. Also, comparing with the variance definition of [Zitong Yang, 2020], have you tried multiple loss functions and observe the same phenomenon?

- minor issue: the sample-wise error for one example x should be explicitly listed and the notation should be consistent through the whole paper, y(x), and y should not be mixed together. I know the original [Heskes 1998] paper did the same, but you should be consistent on this notation.

- other minor things: Could you please elaborate a bit more of how you preprocess the imagenet data into 1.2 million images and how many classes are remained? This piece of information seems missing and the code doesn't reveal much about it.

---

> ### Author Response · Authors · 2020-11-14
> **Author response to AnonReviewer3 (1/2)**
>
> We thank the reviewer for insightful and constructive comments.
>
> ---
> - Q1: My major concern comes from the definition of variance and its corresponding implementation. The definition of variance is flawed. The formulation doesn't match with the figure illustration and code implementation.
> - A1: The variance shown in Fig 1 is to illustrate that the variance caused by __different teachers__ is similar to the variance caused by __different training datasets__. The figure is inspired by the book [1] and a blog post [2]. If we train a model, the chosen training dataset is a sampling from the ground-truth data distribution. Then different choices of training dataset introduce variance. Similarly, in our definition of variance, we think the selection of a teacher network introduces randomness. With different teacher networks, we will have different soft labels, so the final performance will be different.
> Our implementation is to investigate the sample-wise bias-variance caused by distillation. The corresponding implementation is like the practice of tuning the model complexity we use in machine learning. For example, given a training dataset, we need to decide to use a linear regression model or a quadric regression. Since increasing model complexity leads to a larger variance, the quadric regression has a larger variance than the linear regression. Note that in this process, we do not estimate the variance by using multiple training datasets. This is because the variance describes how much the model predictions will change if the training set changes. In our paper, bias and variance are also decomposed from the generalization error, and decrease one will increase another. __So in our implementation, we do not estimate the variance__, which is similar to choosing a linear regression model or a quadric regression for a machine learning task. __Overall, our code implementation matches our algorithm and problem formulation.__
>
> We revised Figure 1 to avoid misleading presentations. The teachers in the figure are changed to soft label sets, which are more consistent with the variance illustrated in [1] and [2]. Finally, thanks for this helpful comment.
>
> [1] Friedman, Jerome, Trevor Hastie, and Robert Tibshirani. The elements of statistical learning. https://web.stanford.edu/~hastie/ElemStatLearn/printings/ESLII_print12_toc.pdf (PDF Page 244 Figure 7.2)
> [2] https://blog.insightdatascience.com/bias-variance-tradeoff-explained-fa2bc28174c4 (the first figure)
>
> ---
> - Q2: Second concern is that the mathematical reasons behind the not-too-big or not-too-small weights for these regularization examples are unclear.
> - A2: Our weighting scheme is based on a straightforward motivation: We need to adaptively handle the regularization samples and the weight should be in $[0,1]$. As mentioned in our paper, the weighting form is inspired by previous practice that verified useful (Lin et al., 2017; Tang et al., 2019). In fact, we have tried other straightforward weighting schemes. Since the weight should be a number between $[0,1]$ and $\frac{L_{ce}^s}{L_{ce}^t}$ is the input, a baseline idea is that we can use sigmoid to do this. Note that $\frac{L_{ce}^s}{L_{ce}^t}$ is always bigger than 0, so the weight needs scaling and can be defined as $\frac{2}{1+\exp(-\frac{L_{ce}^s}{L_{ce}^t})}-1$. The comparison between the adopted weighting form and the Sigmoid baseline is (CIFAR 100, WRN-40-2 teach WRN-40-1)
>
> |$\alpha$|2.0|3.0|4.0|
> |---|---|---|---|
> |Sigmoid baseline|74.13|73.97|73.29|
> |Ours|74.38|74.12|73.46|
>
> We can see that as long as we can adaptively tune the sample-wise bias-variance tradeoff, the performance is better than KD, i.e., without weighted soft labels. Therefore, although the proposed weighting form is not mathematically optimal, the not-too-big or not-too-small weights for these regularization examples are not hard to tune. These results verify our main contribution that there is a sample-wise bias-variance tradeoff and we need to assign weights to the regularization examples. The results and discussions are updated in the appendix.
>
> ---
> - Q3: The relation (similarity and difference) between your employed BVD from [Heskes 1998] and [Pedro Domingos 2002] is not discussed. [Giorgio Valentini 2004] is not even cited, which seems inappropriate. Also, comparing with the variance definition of [Zitong Yang, 2020], have you tried multiple loss functions and observe the same phenomenon?
> - A3: Thanks for the closely related work. Our employed BVD is directly from the results in [Heskes 1998]. We add new words in the paper to emphasize the relation and include [Giorgio Valentini 2004] as well. We did not try other loss functions, since the scope of this paper is to investigate knowledge distillation, which is defined by KL divergence. We agree that trying other loss functions is an important future direction of this paper.

---

> > ### Author Response · Authors · 2020-11-14
> > **Author response to AnonReviewer3 (2/2)**
> >
> > - Q4: The expectation's definition for Equation 1 is not clear. The notation should be consistent through the whole paper.
> > - A4: Really thanks for pointing this out. We have revised the section to avoid the abuse of notations. A new draft is uploaded where text in brown is the revised part. Please let us know if there are still notation mistakes.
> >
> > ---
> > - Q5: Could you please elaborate a bit more of how you preprocess the imagenet data into 1.2 million images and how many classes are remained?
> > - A5: We use the full imagenet dataset. After downloading it from the official ImageNet website, we compact it into LMDB format. All images are used, so in total, we will have 1.2 million images and 1000 classes. We preprocess the imagenet dataset exactly the same as all comparison methods.
> >
> > ---
> > - Q6: The observed phenomenon regarding the regularization examples is not well explained.
> > - A6: We are not sure if our replies to the above questions covered this point. Please let us know if there is still not well-explained phenomenon regarding the regularization examples.

---

### Official Review · AnonReviewer1 · 2020-10-30
**A new perspective to understand the effect of soft labels in KD**

**Rating:** 7
**Confidence:** 4

**Review:**

This paper provides a way to research the effect of soft labels in knowledge distillation from the perspective of sample-wise bias-variance tradeoff.  Based on some observations about regularization samples, the authors propose the weighted soft labels to handle the tradeoff.  Experiments on standard datasets show that the proposed method can improve the standard knowledge distillation.

The motivation and logic of the article are clear. Based on the form of rewriting distillation loss into a regularization loss adding the direct training loss, this work introduces the sample-wise bias-variance tradeoff, which help to understand the regularization effect of KD.  Despite the weighted soft label method is heuristic, given the simplicity and effectiveness of the proposed method, I think this paper is satisfactory.

Still, I would like to give the authors some suggestions:
1.Section 3.4 should do more experiments about the intermediate state between excluding regularization samples and only on regularization samples to clearly show the tradeoff.
2.Can the authors give some quantitative results in the Table 5 experiments to show how weighted soft labels tune sample-wise bias-variance?
3.Can the weighted soft labels be applied to the variants of KD?  It would be better to show some results about this respect.

---

> ### Author Response · Authors · 2020-11-14
> **Author response to AnonReviewer1**
>
> We thank the reviewer for his encouraging feedback and appreciation of our work.
>
> ---
> - Q1: Section 3.4 should do more experiments about the intermediate state between excluding regularization samples and only on regularization samples to clearly show the tradeoff.
> - A1: Thanks for this suggestion. We add the following experiments: 1) Gradually excluding regularization samples and 2) Gradually adding non-regularization samples while regularization samples are kept. For adding or excluding a sample, we set a probability to achieve this goal. For example, if during training, a sample is marked as a regularization sample according to the value of $a$ and $b$, we backward the loss of this sample by a probability $p$. In this way, we can gradually change the training settings. Specifically, the setting is teaching WRN-40-1 with WRN-40-2 and the results are as follows:
>
> |excluding all regularization|excluding 0.75 regularization|excluding 0.50 regularization|excluding 0.25 regularization|
> |---|---|---|---|
> |74.59|74.63|74.72|74.87|
>
> The other results are
>
> |only on regularization|adding 0.25 non-regularization|adding 0.50 non-regularization|add 0.75 non-regularization|
> |---|---|---|---|
> |73.86|74.12|74.47|74.71|
>
> For the above tables, the values after excluding or adding are the probabilities of excluding or adding this sample. The results further verify our motivation that we need to balance the weights between non-regularization and regularization during training. These results are updated in the appendix.
>
> ---
> - Q2: Can the authors give some quantitative results in Table 5 experiments to show how weighted soft labels tune sample-wise bias-variance?
> - A2: Similar to Q1, we conduct experiments with weighted soft labels for the intermediate states.
>
> |excluding all regularization|excluding 0.75 regularization|excluding 0.50 regularization|excluding 0.25 regularization|
> |---|---|---|---|
> |75.35|75.48|75.61|75.72|
>
> and
>
> |only on regularization|adding 0.25 non-regularization|adding 0.50 non-regularization|add 0.75 non-regularization|
> |---|---|---|---|
> |74.46|74.79|75.18|75.53|
>
> Compared with the results presented in Q1, we can observe that weighted soft labels are indeed balancing the sample-wise, not on dataset scale, bias, and variance. The balancing is achieved by lower the weights of regularization samples during training, instead of dropping them. Please let us know if further quantitative results are needed.
>
> ---
> - Q3: Can the weighted soft labels be applied to the variants of KD? It would be better to show some results in this respect.
> - A3: Thanks for the suggestion. We implement the setting WSL+RKD [1]. The results on CIFAR-100 are as follows (averaged over 5 runs):
>
> |Distillation setting|WRN-40-2$\rightarrow$WRN-16-2|WRN-40-2$\rightarrow$WRN-40-1|resnet56$\rightarrow$resnet20|
> |---|---|---|---|
> |Teacher |75.61|75.61|72.34|
> |Student |73.26|71.98|69.06|
> |RDK |74.12|73.34|70.25|
> |WSL + RKD |74.65|73.89|70.73|
>
> We can observe that the weighted soft label applied to RKD still brings improvements, though not that big compared with that applied to KD. We believe that it is an important future direction to explore the applications to more variants of KD.
>
> [1] Wonpyo Park, Dongju Kim, Yan Lu, and Minsu Cho.  Relational knowledge distillation.

---

### Author Response · Authors · 2020-11-21
**Submission revision -- summary of changes**

We thank all the reviewers for their constructive comments. We have revised our paper accordingly and the revised parts are highlighted as brown. Specifically, we have made the following changes:

1. We revise our mathematical definitions to avoid abuse of notations.
2. We add experiments about the intermediate state between excluding regularization samples and only on regularization samples to clearly show the tradeoff.
3. We add experiments about WSL+RKD to investigate how the weighted soft labels can be applied to the variants of KD.
4. We add experiments about other balancing variants.
5. We add experiments on an NLP task.
6. We add experiments about the impact of the hyperparameter $\alpha$.

---

### Decision · Program_Chairs · 2021-01-07
**Final Decision**

**Decision:**

Accept (Poster)

**Comment:**

The paper investigates the effect of soft labels in knowledge distillation from the perspective of sample-wise bias-variance tradeoff. They observe that during training the bias-variance tradeoff
varies sample-wisely. and under the same distillation temperature setting, we
 distillation performance is negatively associated with the number of regularization samples. But removing them altogether hurts the performance (the authors show empirical evidence of this). Based on some observations about regularization samples, the authors propose the weighted soft labels to handle the tradeoff. Experiments on standard datasets show that the proposed method can improve the standard knowledge distillation.

pros.
-the paper is written clearly.
-through the review period the authors added additional experiments suggested by the reviewers and enhances experimental results. The experiment results are convincing and the authors have now added explanations on hyperparameter choices.
-the mathematical setting is now clear after incorporating reviewer's comments.
-the missing related work as suggested by reviewers is added

cons.
-comparison with results of Zitong Yang et al 2020[1] is missing.

I thank the authors for incorporating the changes requested by reviewers. Please add comparison with result of [1] in the final version.

[1] Rethinking Bias-Variance Trade-off for Generalization of Neural Networks
Zitong Yang, Yaodong Yu, Chong You, Jacob Steinhardt, Yi Ma